# Bayesian GAN

**Yunus Saatchi**
Uber AI Labs

**Andrew Gordon Wilson**
Cornell University

## Abstract

Generative adversarial networks (GANs) can implicitly learn rich distributions over images, audio, and data which are hard to model with an explicit likelihood. We present a practical Bayesian formulation for unsupervised and semi-supervised learning with GANs. Within this framework, we use stochastic gradient Hamiltonian Monte Carlo to marginalize the weights of the generator and discriminator networks. The resulting approach is straightforward and obtains good performance without any standard interventions such as label smoothing or mini-batch discrimination. By exploring an expressive posterior over the parameters of the generator, the Bayesian GAN avoids mode-collapse, produces interpretable and diverse candidate samples, and provides state-of-the-art quantitative results for semi-supervised learning on benchmarks including SVHN, CelebA, and CIFAR-10, outperforming DCGAN, Wasserstein GANs, and DCGAN ensembles.

## 1  Introduction

Learning a good generative model for high-dimensional natural signals, such as images, video and audio has long been one of the key milestones of machine learning. Powered by the learning capabilities of deep neural networks, generative adversarial networks (GANs) [4] and variational autoencoders [6] have brought the field closer to attaining this goal.

GANs transform white noise through a deep neural network to generate candidate samples from a data distribution. A discriminator learns, in a supervised manner, how to tune its parameters so as to correctly classify whether a given sample has come from the generator or the true data distribution. Meanwhile, the generator updates its parameters so as to fool the discriminator. As long as the generator has sufficient capacity, it can approximate the CDF inverse-CDF composition required to sample from a data distribution of interest. Since convolutional neural networks by design provide reasonable metrics over images (unlike, for instance, Gaussian likelihoods), GANs using convolutional neural networks can in turn provide a compelling implicit distribution over images.

Although GANs have been highly impactful, their learning objective can lead to *mode collapse*, where the generator simply memorizes a few training examples to fool the discriminator. This pathology is reminiscent of maximum likelihood density estimation with Gaussian mixtures: by collapsing the variance of each component we achieve infinite likelihood and memorize the dataset, which is not useful for a generalizable density estimate. Moreover, a large degree of intervention is required to stabilize GAN training, including label smoothing and mini-batch discrimination [9, 10]. To help alleviate these practical difficulties, recent work has focused on replacing the Jensen-Shannon divergence implicit in standard GAN training with alternative metrics, such as f-divergences [8] or Wasserstein divergences [1]. Much of this work is analogous to introducing various regularizers for maximum likelihood density estimation. But just as it can be difficult to choose the *right* regularizer, it can also be difficult to decide which divergence we wish to use for GAN training.

It is our contention that GANs can be improved by fully probabilistic inference. Indeed, a posterior distribution over the parameters of the generator could be broad and highly multimodal. GAN training, which is based on mini-max optimization, always estimates this whole posterior distribution

over the network weights as a point mass centred on a single mode. Thus even if the generator does *not* memorize training examples, we would expect samples from the generator to be overly compact relative to samples from the data distribution. Moreover, each mode in the posterior over the network weights could correspond to wildly different generators, each with their own meaningful interpretations. By fully representing the posterior distribution over the parameters of both the generator and discriminator, we can more accurately model the true data distribution. The inferred data distribution can then be used for accurate and highly data-efficient *semi-supervised* learning.

In this paper, we propose a simple Bayesian formulation for end-to-end unsupervised and semi-supervised learning with generative adversarial networks. Within this framework, we marginalize the posteriors over the weights of the generator and discriminator using stochastic gradient Hamiltonian Monte Carlo. We interpret data samples from the generator, showing exploration across several distinct modes in the generator weights. We also show data and iteration efficient learning of the true distribution. We also demonstrate state of the art semi-supervised learning performance on several benchmarks, including SVHN, MNIST, CIFAR-10, and CelebA. The simplicity of the proposed approach is one of its greatest strengths: inference is straightforward, interpretable, and stable. Indeed all of the experimental results were obtained without many of the ad-hoc techniques often used to train standard GANs.

We have made code and tutorials available at
https://github.com/andrewgordonwilson/bayesgan.

## 2 Bayesian GANs

Given a dataset $\mathcal{D} = \{\mathbf{x}^{(i)}\}$ of variables $\mathbf{x}^{(i)} \sim p_{\text{data}}(\mathbf{x}^{(i)})$, we wish to estimate $p_{\text{data}}(\mathbf{x})$. We transform white noise $\mathbf{z} \sim p(\mathbf{z})$ through a generator $G(\mathbf{z}; \theta_g)$, parametrized by $\theta_g$, to produce candidate samples from the data distribution. We use a discriminator $D(\mathbf{x}; \theta_d)$, parametrized by $\theta_d$, to output the probability that any $\mathbf{x}$ comes from the data distribution. Our considerations hold for general $G$ and $D$, but in practice $G$ and $D$ are often neural networks with weight vectors $\theta_g$ and $\theta_d$.

By placing distributions over $\theta_g$ and $\theta_d$, we induce distributions over an uncountably infinite space of generators and discriminators, corresponding to every possible setting of these weight vectors. The generator now represents a *distribution over distributions* of data. Sampling from the induced prior distribution over data instances proceeds as follows:

(1) Sample $\theta_g \sim p(\theta_g)$; (2) Sample $\mathbf{z}^{(1)}, \ldots, \mathbf{z}^{(n)} \sim p(\mathbf{z})$; (3) $\tilde{\mathbf{x}}^{(j)} = G(\mathbf{z}^{(j)}; \theta_g) \sim p_{\text{generator}}(\mathbf{x})$. For posterior inference, we propose unsupervised and semi-supervised formulations in Sec 2.1 - 2.2.

We note that in an exciting recent work Tran et al. [11] briefly mention using a variational approach to marginalize weights in a generative model, as part of a general exposition on hierarchical implicit models (see also Karaletsos [5] for a nice theoretical exploration of related topics in graphical model message passing). While this related work is promising, our approach has several key differences: (1) our GAN representation is quite different, with novel formulations for the conditional posteriors, preserving a clear competition between generator and discriminator; (2) our representation for the posteriors is straightforward, provides novel formulations for unsupervised and semi-supervised learning, and has state of the art results on many benchmarks. Conversely, Tran et al. [11] is only pursued for fully supervised learning on a few small datasets; (3) we use sampling to explore a full posterior over the weights, whereas Tran et al. [11] perform a variational approximation centred on one of the modes of the posterior (and due to the properties of the KL divergence is prone to an overly compact representation of even that mode); (4) we marginalize $\mathbf{z}$ in addition to $\theta_g, \theta_d$; and (5) the ratio estimation approach in [11] limits the size of the neural networks they can use, whereas in our experiments we can use comparably deep networks to maximum likelihood approaches. In the experiments we illustrate the practical value of our formulation.

Although the high level concept of a Bayesian GAN has been informally mentioned in various contexts, to the best of our knowledge we present the first detailed treatment of Bayesian GANs, including novel formulations, sampling based inference, and rigorous semi-supervised learning experiments.

## 2.1 Unsupervised Learning

To infer posteriors over $\theta_g$, $\theta_d$, we propose to iteratively sample from the following conditional posteriors:

$$p(\theta_g | \mathbf{z}, \theta_d) \propto \left( \prod_{i=1}^{n_g} D(G(\mathbf{z}^{(i)}; \theta_g); \theta_d) \right) p(\theta_g | \alpha_g) \tag{1}$$

$$p(\theta_d | \mathbf{z}, \mathbf{X}, \theta_g) \propto \prod_{i=1}^{n_d} D(\mathbf{x}^{(i)}; \theta_d) \times \prod_{i=1}^{n_g} (1 - D(G(\mathbf{z}^{(i)}; \theta_g); \theta_d)) \times p(\theta_d | \alpha_d). \tag{2}$$

$p(\theta_g | \alpha_g)$ and $p(\theta_d | \alpha_d)$ are priors over the parameters of the generator and discriminator, with hyperparameters $\alpha_g$ and $\alpha_d$, respectively. $n_d$ and $n_g$ are the numbers of mini-batch samples for the discriminator and generator, respectively.[1] We define $\mathbf{X} = \{\mathbf{x}^{(i)}\}_{i=1}^{n_d}$.

We can intuitively understand this formulation starting from the generative process for data samples. Suppose we were to sample weights $\theta_g$ from the prior $p(\theta_g | \alpha_g)$, and then condition on this sample of the weights to form a particular generative neural network. We then sample white noise $\mathbf{z}$ from $p(\mathbf{z})$, and transform this noise through the network $G(\mathbf{z}; \theta_g)$ to generate candidate data samples. The discriminator, conditioned on its weights $\theta_d$, outputs a probability that these candidate samples came from the data distribution. Eq. (1) says that if the discriminator outputs high probabilities, then the posterior $p(\theta_g | \mathbf{z}, \theta_d)$ will increase in a neighbourhood of the sampled setting of $\theta_g$ (and hence decrease for other settings). For the posterior over the discriminator weights $\theta_d$, the first two terms of Eq. (2) form a discriminative classification likelihood, labelling samples from the actual data versus the generator as belonging to separate classes. And the last term is the prior on $\theta_d$.

**Classical GANs as maximum likelihood**   Moreover, our proposed probabilistic approach is a natural Bayesian generalization of the classical GAN: if one uses uniform priors for $\theta_g$ and $\theta_d$, and performs iterative MAP optimization instead of posterior sampling over Eq. (1) and (2), then the local optima will be the same as for Algorithm 1 of Goodfellow et al. [4]. We thus sometimes refer to the classical GAN as the ML-GAN. Moreover, even with a flat prior, there is a big difference between Bayesian marginalization over the whole posterior versus approximating this (often broad, multimodal) posterior with a point mass as in MAP optimization (see Figure 3, Supplement).

**Marginalizing the noise**   In prior work, GAN updates are implicitly conditioned on a set of noise samples $\mathbf{z}$. We can instead marginalize $\mathbf{z}$ from our posterior updates using simple Monte Carlo:

$$p(\theta_g | \theta_d) = \int p(\theta_g, \mathbf{z} | \theta_d) d\mathbf{z} = \int p(\theta_g | \mathbf{z}, \theta_d) \overbrace{p(\mathbf{z} | \theta_d)}^{=p(\mathbf{z})} d\mathbf{z} \approx \frac{1}{J_g} \sum_{j=1}^{J_g} p(\theta_g | \mathbf{z}^{(j)}, \theta_d), \; \mathbf{z}^{(j)} \sim p(\mathbf{z})$$

By following a similar derivation, $p(\theta_d | \theta_g) \approx \frac{1}{J_d} \sum_{j}^{J_d} p(\theta_d | \mathbf{z}^{(j)}, \mathbf{X}, \theta_g)$, $\mathbf{z}^{(j)} \sim p(\mathbf{z})$.

This specific setup has several nice features for Monte Carlo integration. First, $p(\mathbf{z})$ is a white noise distribution from which we can take efficient and exact samples. Secondly, both $p(\theta_g | \mathbf{z}, \theta_d)$ and $p(\theta_d | \mathbf{z}, \mathbf{X}, \theta_g)$, when viewed as a function of $\mathbf{z}$, should be reasonably broad over $\mathbf{z}$ by construction, since $\mathbf{z}$ is used to produce candidate data samples in the generative procedure. Thus each term in the simple Monte Carlo sum typically makes a reasonable contribution to the total marginal posterior estimates. We do note, however, that the approximation will typically be worse for $p(\theta_d | \theta_g)$ due to the conditioning on a minibatch of data in Equation 2.

**Posterior samples**   By iteratively sampling from $p(\theta_g | \theta_d)$ and $p(\theta_d | \theta_g)$ at every step of an epoch one can, in the limit, obtain samples from the approximate posteriors over $\theta_g$ and $\theta_d$. Having such samples can be very useful in practice. Indeed, one can use different samples for $\theta_g$ to alleviate GAN collapse and generate data samples with an appropriate level of entropy, as well as forming a committee of generators to strengthen the discriminator. The samples for $\theta_d$ in turn form a committee of discriminators which amplifies the overall adversarial signal, thereby further improving the unsupervised learning process. Arguably, the most rigorous method to assess the utility of these posterior samples is to examine their effect on *semi-supervised* learning, which is a focus of our experiments in Section 4.

## 2.2 Semi-supervised Learning

We extend the proposed probabilistic GAN formalism to semi-supervised learning. In the semi-supervised setting for $K$-class classification, we have access to a set of $n$ unlabelled observations, $\{\mathbf{x}^{(i)}\}$, as well as a (typically much smaller) set of $n_s$ observations, $\{(\mathbf{x}_s^{(i)}, y_s^{(i)})\}_{i=1}^{N_s}$, with class labels $y_s^{(i)} \in \{1, \dots, K\}$. Our goal is to jointly learn statistical structure from both the unlabelled and labelled examples, in order to make much better predictions of class labels for new test examples $\mathbf{x}_*$ than if we only had access to the labelled training inputs.

In this context, we redefine the discriminator such that $D(\mathbf{x}^{(i)} = y^{(i)}; \theta_d)$ gives the probability that sample $\mathbf{x}^{(i)}$ belongs to class $y^{(i)}$. We reserve the class label 0 to indicate that a data sample is the output of the generator. We then infer the posterior over the weights as follows:

$$p(\theta_g | \mathbf{z}, \theta_d) \propto \left( \prod_{i=1}^{n_g} \sum_{y=1}^{K} D(G(\mathbf{z}^{(i)}; \theta_g) = y; \theta_d) \right) p(\theta_g | \alpha_g) \tag{3}$$

$$p(\theta_d | \mathbf{z}, \mathbf{X}, \mathbf{y}_s, \theta_g) \propto \prod_{i=1}^{n_d} \sum_{y=1}^{K} D(\mathbf{x}^{(i)} = y; \theta_d) \prod_{i=1}^{n_g} D(G(\mathbf{z}^{(i)}; \theta_g) = 0; \theta_d) \prod_{i=1}^{N_s} (D(\mathbf{x}_s^{(i)} = y_s^{(i)}; \theta_d)) p(\theta_d | \alpha_d)$$
$$\tag{4}$$

During every iteration we use $n_g$ samples from the generator, $n_d$ *unlabeled* samples, and all of the $N_s$ labeled samples, where typically $N_s \ll n$. As in Section 2.1, we can approximately marginalize $\mathbf{z}$ using simple Monte Carlo sampling.

Much like in the unsupervised learning case, we can marginalize the posteriors over $\theta_g$ and $\theta_d$. To compute the predictive distribution for a class label $y_*$ at a test input $\mathbf{x}_*$ we use a model average over all collected samples with respect to the posterior over $\theta_d$:

$$p(y_* | \mathbf{x}_*, \mathcal{D}) = \int p(y_* | \mathbf{x}_*, \theta_d) p(\theta_d | \mathcal{D}) d\theta_d \approx \frac{1}{T} \sum_{k=1}^{T} p(y_* | \mathbf{x}_*, \theta_d^{(k)}), \theta_d^{(k)} \sim p(\theta_d | \mathcal{D}). \tag{5}$$

We will see that this model average is effective for boosting semi-supervised learning performance. In Section 3 we present an approach to MCMC sampling from the posteriors over $\theta_g$ and $\theta_d$.

## 3  Posterior Sampling with Stochastic Gradient HMC

In the Bayesian GAN, we wish to marginalize the posterior distributions over the generator and discriminator weights, for unsupervised learning in 2.1 and semi-supervised learning in 2.2. For this purpose, we use Stochastic Gradient Hamiltonian Monte Carlo (SGHMC) [3] for posterior sampling. The reason for this choice is three-fold: (1) SGHMC is very closely related to momentum-based SGD, which we know empirically works well for GAN training; (2) we can import parameter settings (such as learning rates and momentum terms) from SGD directly into SGHMC; and most importantly, (3) many of the practical benefits of a Bayesian approach to GAN inference come from exploring a rich multimodal distribution over the weights $\theta_g$ of the generator, which is enabled by SGHMC. Alternatives, such as variational approximations, will typically centre their mass around a single mode, and thus provide a unimodal and otherwise compact representation for the distribution, due to asymmetric biases of the KL-divergence.

The posteriors in Equations 3 and 4 are both amenable to HMC techniques as we can compute the gradients of the loss with respect to the parameters we are sampling. SGHMC extends HMC to the case where we use noisy estimates of such gradients in a manner which guarantees mixing in the limit of a large number of minibatches. For a detailed review of SGHMC, please see Chen et al. [3]. Using the update rules from Eq. (15) in Chen et al. [3], we propose to sample from the posteriors over the generator and discriminator weights as in Algorithm 1. Note that Algorithm 1 outlines standard momentum-based SGHMC: in practice, we found it helpful to speed up the "burn-in" process by replacing the SGD part of this algorithm with Adam for the first few thousand iterations, after which we revert back to momentum-based SGHMC. As suggested in Appendix G of Chen et al. [3], we employed a learning rate schedule which decayed according to $\gamma/d$ where $d$ is set to the number of unique "real" datapoints seen so far. Thus, our learning rate schedule converges to $\gamma/N$ in the limit, where we have defined $N = |\mathcal{D}|$.

**Algorithm 1** One iteration of sampling for the Bayesian GAN. $\alpha$ is the friction term for SGHMC, $\eta$ is the learning rate. We assume that the stochastic gradient discretization noise term $\hat{\beta}$ is dominated by the main friction term (this assumption constrains us to use small step sizes). We take $J_g$ and $J_d$ simple MC samples for the generator and discriminator respectively, and $M$ SGHMC samples for each simple MC sample. We rescale to accommodate minibatches as in the supplementary material.

---

• Represent posteriors with samples $\{\theta_g^{j,m}\}_{j=1,m=1}^{J_g,M}$ and $\{\theta_d^{j,m}\}_{j=1,m=1}^{J_d,M}$ from previous iteration
**for** number of MC iterations $J_g$ **do**
    • Sample $J_g$ noise samples $\{\mathbf{z}^{(1)},\ldots,\mathbf{z}^{(J_g)}\}$ from noise prior $p(\mathbf{z})$. Each $\mathbf{z}^{(i)}$ has $n_g$ samples.
    • Update sample set representing $p(\theta_g|\theta_d)$ by running SGHMC updates for $M$ iterations:

$$\theta_g^{j,m} \leftarrow \theta_g^{j,m} + \mathbf{v}; \ \mathbf{v} \leftarrow (1-\alpha)\mathbf{v} + \eta \frac{\partial \log\left(\sum_i \sum_k p(\theta_g|\mathbf{z}^{(i)},\theta_d^{k,m})\right)}{\partial \theta_g} + \mathbf{n}; \ \mathbf{n} \sim \mathcal{N}(0, 2\alpha\eta I)$$

    • Append $\theta_g^{j,m}$ to sample set.
**end for**
**for** number of MC iterations $J_d$ **do**
    • Sample minibatch of $J_d$ noise samples $\{\mathbf{z}^{(1)},\ldots,\mathbf{z}^{(J_d)}\}$ from noise prior $p(\mathbf{z})$.
    • Sample minibatch of $n_d$ data samples $\mathbf{x}$.
    • Update sample set representing $p(\theta_d|\mathbf{z},\theta_g)$ by running SGHMC updates for $M$ iterations:

$$\theta_d^{j,m} \leftarrow \theta_d^{j,m} + \mathbf{v}; \ \mathbf{v} \leftarrow (1-\alpha)\mathbf{v} + \eta \frac{\partial \log\left(\sum_i \sum_k p(\theta_d|\mathbf{z}^{(i)},\mathbf{x},\theta_g^{k,m})\right)}{\partial \theta_d} + \mathbf{n}; \ \mathbf{n} \sim \mathcal{N}(0, 2\alpha\eta I)$$

    • Append $\theta_d^{j,m}$ to sample set.
**end for**

---

# 4 Experiments

We evaluate our proposed Bayesian GAN (henceforth titled BayesGAN) on six benchmarks (synthetic, MNIST, CIFAR-10, SVHN, and CelebA) each with four different numbers of labelled examples. We consider multiple alternatives, including: the DCGAN [9], the recent Wasserstein GAN (W-DCGAN) [1], an ensemble of ten DCGANs (DCGAN-10) which are formed by 10 random subsets 80% the size of the training set, and a fully supervised convolutional neural network. We also compare to the reported MNIST result for the LFVI-GAN, briefly mentioned in a recent work [11], where they use fully supervised modelling on the whole dataset with a variational approximation. We interpret many of the results from MNIST in detail in Section 4.2, and find that these observations carry forward to our CIFAR-10, SVHN, and CelebA experiments.

For all real experiments except MNIST we use a 6-layer Bayesian deconvolutional GAN (BayesGAN) for the generative model $G(\mathbf{z}|\theta_g)$ (see Radford et al. [9] for further details about structure). The corresponding discriminator is a 6-layer 2-class DCGAN for the unsupervised GAN and a 6-layer, $K+1$ class DCGAN for a semi-supervised GAN performing classification over $K$ classes. The connectivity structure of the unsupervised and semi-supervised DCGANs were the same as for the BayesGAN. As recommended by [10], we used feature matching for all models on semi-supervised experiments. For MNIST we found that 4-layers for all networks worked slightly better across the board, due to the added simplicity of the dataset. Note that the structure of the networks in Radford et al. [9] are slightly different from [10] (e.g. there are 4 hidden layers and fewer filters per layer), so one cannot directly compare the results here with those in Salimans et al. [10]. Our exact architecture specification is also given in our codebase. The performance of all methods could be improved through further calibrating architecture design for each individual benchmark. For the Bayesian GAN we place a $\mathcal{N}(0, 10I)$ prior on both the generator and discriminator weights and approximately integrate out $\mathbf{z}$ using simple Monte Carlo samples. We run Algorithm 1 for 5000 iterations and then collect weight samples every 1000 iterations and record out-of-sample predictive accuracy using Bayesian model averaging (see Eq. 5). For Algorithm 1 we set $J_g = 10$, $J_d = 1$, $M = 2$, and $n_d = n_g = 64$. All experiments were performed on a single TitanX GPU for consistency, but BayesGAN and DCGAN-10 could be sped up to approximately the same runtime as DCGAN through multi-GPU parallelization.

In Table 1 we summarize the semi-supervised results, where we see consistently improved performance over the alternatives. All runs are averaged over 10 random subsets of labeled examples for estimating error bars on performance (Table 1 shows mean and 2 standard deviations). We also qualitatively illustrate the ability for the Bayesian GAN to produce complementary sets of data samples, corresponding to different representations of the generator produced by sampling from the posterior over the generator weights (Figures 1, 2, 5). The supplement also contains additional plots of accuracy per epoch for semi-supervised experiments.

## 4.1 Synthetic Dataset

We present experiments on a multi-modal synthetic dataset to test the ability to infer a multi-modal posterior $p(\theta_g|\mathcal{D})$. This ability not only helps avoid the collapse of the generator to a couple training examples, an instance of overfitting in regular GAN training, but also allows one to explore a set of generators with different complementary properties, harmonizing together to encapsulate a rich data distribution. We generate $D$-dimensional synthetic data as follows:

$$\mathbf{z} \sim \mathcal{N}(0, 10 * I_d), \quad \mathbf{A} \sim \mathcal{N}(0, I_{D \times d}), \quad \mathbf{x} = \mathbf{A}\mathbf{z} + \epsilon, \quad \epsilon \sim \mathcal{N}(0, 0.01 * I_D), \quad d \ll D$$

We then fit both a regular GAN and a Bayesian GAN to such a dataset with $D = 100$ and $d = 2$. The generator for both models is a two-layer neural network: 10-1000-100, fully connected, with ReLU activations. We set the dimensionality of $\mathbf{z}$ to be 10 in order for the DCGAN to converge (it does not converge when $d = 2$, despite the inherent dimensionality being 2!). Consistently, the discriminator network has the following structure: 100-1000-1, fully-connected, ReLU activations. For this dataset we place an $\mathcal{N}(0, I)$ prior on the weights of the Bayesian GAN and approximately integrate out $\mathbf{z}$ using $J = 100$ Monte-Carlo samples. Figure 1 shows that the Bayesian GAN does a much better job qualitatively in generating samples (for which we show the first two principal components), and quantitatively in terms of Jensen-Shannon divergence (JSD) to the true distribution (determined through kernel density estimates). In fact, the DCGAN (labelled ML GAN as per Section 2) begins to eventually increase in testing JSD after a certain number of training iterations, which is reminiscent of over-fitting. When $D = 500$, we still see good performance with the Bayesian GAN. We also see, with multidimensional scaling [2], that samples from the posterior over Bayesian generator weights clearly form multiple distinct clusters, indicating that the SGHMC sampling is exploring multiple distinct modes, thus capturing multimodality in *weight space* as well as in data space.

## 4.2 MNIST

MNIST is a well-understood benchmark dataset consisting of 60k (50k train, 10k test) labeled images of hand-written digits. Salimans et al. [10] showed excellent out-of-sample performance using only a small number of labeled inputs, convincingly demonstrating the importance of good generative modelling for semi-supervised learning. Here, we follow their experimental setup for MNIST.

We evaluate the Bayesian DCGAN for semi-supervised learning using $N_s = \{20, 50, 100, 200\}$ labelled training examples. We see in Table 1 that the Bayesian GAN has improved accuracy over the DCGAN, the Wasserstein GAN, and even an ensemble of 10 DCGANs! Moreover, it is remarkable that the Bayesian GAN with only 100 labelled training examples ($0.2\%$ of the training data) is able to achieve $99.3\%$ testing accuracy, which is comparable with a state of the art fully supervised method using *all* $50,000$ *training examples*! We show a fully supervised model using $n_s$ samples to generally highlight the practical utility of semi-supervised learning.

Moreover, Tran et al. [11] showed that a fully supervised LFVI-GAN, on the whole MNIST training set ($50,000$ labelled examples) produces 140 classification errors – almost twice the error of our proposed Bayesian GAN approach using only $n_s = 100$ ($0.2\%$) labelled examples! We suspect this difference largely comes from (1) the simple practical formulation of the Bayesian GAN in Section 2, (2) marginalizing $\mathbf{z}$ via simple Monte Carlo, and (3) exploring a broad multimodal posterior distribution over the generator weights with SGHMC with our approach versus a variational approximation (prone to over-compact representations) centred on a single mode.

We can also see qualitative differences in the unsupervised data samples from our Bayesian DCGAN and the standard DCGAN in Figure 2. The top row shows sample images produced from *six* generators produced from six samples over the posterior of the generator weights, and the bottom row shows sample data images from a DCGAN. We can see that each of the six panels in the top row have

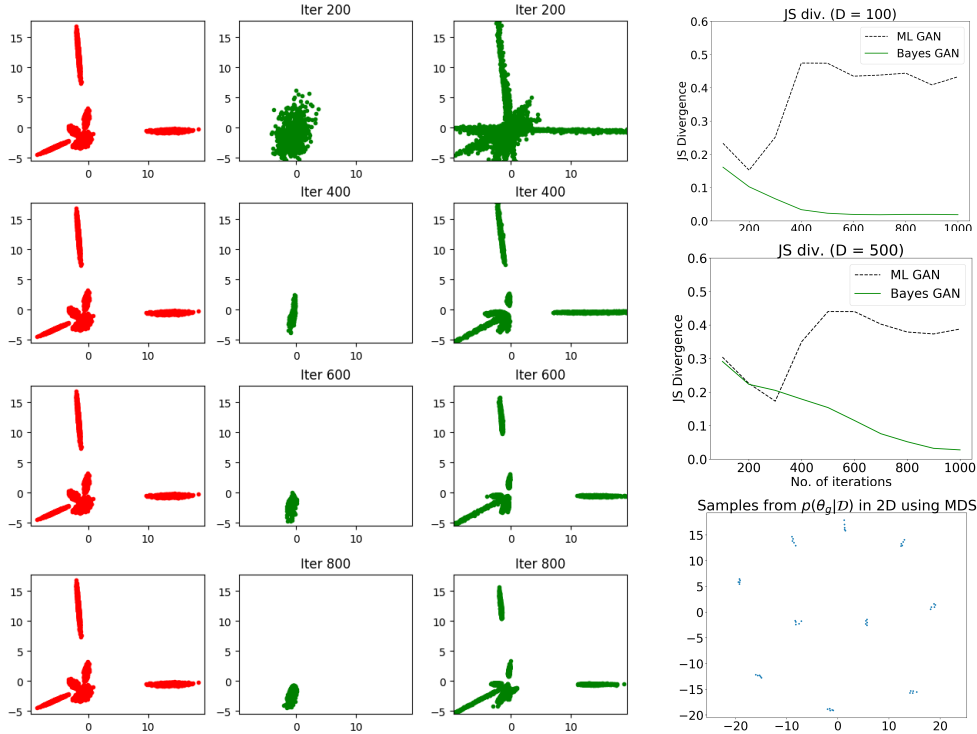

Figure 1: **Left**: Samples drawn from $p_{\text{data}}(\mathbf{x})$ and visualized in 2-D after applying PCA. Right 2 columns: Samples drawn from $p_{\text{MLGAN}}(\mathbf{x})$ and $p_{\text{BGAN}}(\mathbf{x})$ visualized in 2D after applying PCA. The data is inherently 2-dimensional so PCA can explain most of the variance using 2 principal components. It is clear that the Bayesian GAN is capturing all the modes in the data whereas the regular GAN is unable to do so. **Right**: (Top 2) Jensen-Shannon divergence between $p_{\text{data}}(\mathbf{x})$ and $p(\mathbf{x}; \theta)$ as a function of the number of iterations of GAN training for $D = 100$ (top) and $D = 500$ (bottom). The divergence is computed using kernel density estimates of large sample datasets drawn from $p_{\text{data}}(\mathbf{x})$ and $p(\mathbf{x}; \theta)$, after applying dimensionality reduction to 2-D (the inherent dimensionality of the data). In both cases, the Bayesian GAN is far more effective at minimizing the Jensen-Shannon divergence, reaching convergence towards the true distribution, by exploring a full distribution over generator weights, which is not possible with a maximum likelihood GAN (no matter how many iterations). (Bottom) The sample set $\{\theta_g^k\}$ after convergence viewed in 2-D using Multidimensional Scaling (using a Euclidean distance metric between weight samples) [2]. One can clearly see several clusters, meaning that the SGHMC sampling has discovered pronounced modes in the posterior over the weights.

qualitative differences, almost as if a different person were writing the digits in each panel. Panel 1 (top left), for example, is quite crisp, while panel 3 is fairly thick, and panel 6 (top right) has thin and fainter strokes. In other words, the Bayesian GAN is learning different complementary generative hypotheses to explain the data. By contrast, all of the data samples on the bottom row from the DCGAN are homogenous. In effect, each posterior weight sample in the Bayesian GAN corresponds to a different style, while in the standard DCGAN the style is fixed. This difference is further illustrated for all datasets in Figure 5 (supplement). Figure 3 (supplement) also further emphasizes the utility of Bayesian marginalization versus optimization, even with vague priors.

However, we do not necessarily expect high fidelity images from any arbitrary generator sampled from the posterior over generators; in fact, such a generator would probably have less posterior probability than the DCGAN, which we show in Section 2 can be viewed as a maximum likelihood analogue of our approach. The advantage in the Bayesian approach comes from representing a whole space of generators alongside their posterior probabilities.

Practically speaking, we also stress that for reasonable sample generation from the maximum-likelihood DCGAN we had to resort to using tricks including minibatch discrimination, feature normalization and the addition of Gaussian noise to each layer of the discriminator. The Bayesian DCGAN needed none of these tricks. This robustness arises from a Gaussian prior over the weights which provides a useful inductive bias, and due to the MCMC sampling procedure which alleviates

Table 1: Detailed supervised and semi-supervised learning results for all datasets. In almost all experiments BayesGAN outperforms DCGAN and W-DCGAN substantially, and typically even outperforms ensembles of DCGANs. The runtimes, per *epoch*, in minutes, are provided in rows including the dataset name. While all experiments were performed on a single GPU, note that DCGAN-10 and BayesGAN methods can be sped up straightforwardly using multiple GPUs to obtain a similar runtime to DCGAN. Note also that the BayesGAN is generally much more efficient per epoch than the alternatives, as per Figure 4 (supplement). Results are averaged over 10 random supervised subsets $\pm$ 2 stdev. Standard train/test splits are used for MNIST, CIFAR-10 and SVHN. For CelebA we use a test set of size 10k. Test error rates are across the *entire* test set.

| $N_s$ | | No. of misclassifications for MNIST. Test error rate for others. | | | |
|---|---|---|---|---|---|
| | | Supervised | DCGAN | W-DCGAN | DCGAN-10 | BayesGAN |
| MNIST | $N$=50k, $D=(28,28)$ | 16 | 19 | 112 | 39 |
| 20 | | — | $1618 \pm 388$ | $1623 \pm 325$ | $1453 \pm 532$ | $\mathbf{1402 \pm 422}$ |
| 50 | | — | $432 \pm 187$ | $412 \pm 199$ | $329 \pm 139$ | $\mathbf{321 \pm 194}$ |
| 100 | | $2134 \pm 525$ | $121 \pm 18$ | $134 \pm 28$ | $102 \pm 11$ | $\mathbf{98 \pm 13}$ |
| 200 | | $1389 \pm 438$ | $95 \pm 7$ | $91 \pm 10$ | $88 \pm 6$ | $\mathbf{82 \pm 5}$ |
| CIFAR-10 | $N$=50k, $D=(32,32,3)$ | 34 | 38 | 217 | 102 |
| 1000 | | $63.4 \pm 2.6$ | $48.6 \pm 3.4$ | $46.1 \pm 3.6$ | $\mathbf{39.6 \pm 2.8}$ | $41.3 \pm 5.1$ |
| 2000 | | $56.1 \pm 2.1$ | $34.1 \pm 4.1$ | $35.8 \pm 3.8$ | $32.4 \pm 2.9$ | $\mathbf{31.4 \pm 3.6}$ |
| 4000 | | $51.4 \pm 2.9$ | $30.8 \pm 4.6$ | $31.1 \pm 4.7$ | $27.4 \pm 3.2$ | $\mathbf{25.9 \pm 3.7}$ |
| 8000 | | $47.2 \pm 2.2$ | $25.1 \pm 3.3$ | $24.4 \pm 5.5$ | $\mathbf{22.6 \pm 2.2}$ | $23.1 \pm 3.9$ |
| SVHN | $N$=75k, $D=(32,32,3)$ | 31 | 34 | 286 | 107 |
| 500 | | $53.5 \pm 2.5$ | $38.2 \pm 3.1$ | $36.1 \pm 4.2$ | $\mathbf{31.8 \pm 4.1}$ | $32.8 \pm 4.4$ |
| 1000 | | $37.3 \pm 3.1$ | $23.6 \pm 4.6$ | $22.1 \pm 4.8$ | $\mathbf{19.8 \pm 2.1}$ | $21.9 \pm 3.5$ |
| 2000 | | $26.3 \pm 2.1$ | $21.2 \pm 3.1$ | $21.0 \pm 1.3$ | $17.1 \pm 2.3$ | $\mathbf{16.3 \pm 2.4}$ |
| 4000 | | $20.8 \pm 1.8$ | $18.2 \pm 1.7$ | $17.1 \pm 1.2$ | $13.0 \pm 1.9$ | $\mathbf{12.7 \pm 1.4}$ |
| CelebA | $N$=100k, $D=(50,50,3)$ | 109 | 117 | 767 | 387 |
| 1000 | | $53.8 \pm 4.2$ | $48.1 \pm 4.8$ | $45.5 \pm 5.9$ | $43.3 \pm 5.3$ | $\mathbf{42.4 \pm 6.7}$ |
| 2000 | | $36.7 \pm 3.2$ | $31.1 \pm 3.2$ | $30.1 \pm 3.3$ | $28.2 \pm 1.3$ | $\mathbf{26.8 \pm 4.2}$ |
| 4000 | | $34.3 \pm 3.8$ | $28.3 \pm 3.2$ | $26.0 \pm 2.1$ | $\mathbf{21.3 \pm 1.2}$ | $22.6 \pm 3.7$ |
| 8000 | | $31.1 \pm 4.2$ | $22.5 \pm 1.5$ | $21.0 \pm 1.9$ | $20.1 \pm 1.4$ | $\mathbf{19.4 \pm 3.4}$ |

the risk of collapse and helps explore multiple modes (and uncertainty within each mode). To be balanced, we also stress that in practice the risk of collapse is not fully eliminated – indeed, some samples from $p(\theta_g|\mathcal{D})$ still produce generators that create data samples with too little entropy. In practice, sampling is not immune to becoming trapped in sharply peaked modes. We leave further analysis for future work.

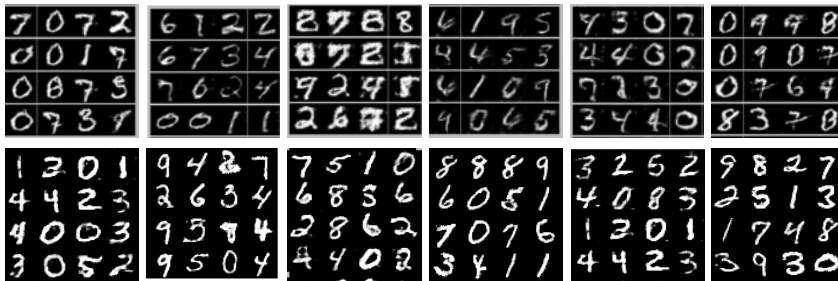

Figure 2: **Top:** Data samples from six different generators corresponding to six samples from the posterior over $\theta_g$. The data samples show that each explored setting of the weights $\theta_g$ produces generators with complementary high-fidelity samples, corresponding to different styles. The amount of variety in the samples emerges naturally using the Bayesian approach. **Bottom:** Data samples from a standard DCGAN (trained six times). By contrast, these samples are homogenous in style.

### 4.3 CIFAR-10

CIFAR-10 is also a popular benchmark dataset [7], with 50k training and 10k test images, which is harder to model than MNIST since the data are 32x32 RGB images of real objects. Figure 5 shows

datasets produced from four different generators corresponding to samples from the posterior over the generator weights. As with MNIST, we see meaningful qualitative variation between the panels. In Table 1 we also see again (but on this more challenging dataset) that using Bayesian GANs as a generative model within the semi-supervised learning setup significantly decreases test set error over the alternatives, especially when $n_s \ll n$.

## 4.4 SVHN

The StreetView House Numbers (SVHN) dataset consists of RGB images of house numbers taken by StreetView vehicles. Unlike MNIST, the digits significantly differ in shape and appearance. The experimental procedure closely followed that for CIFAR-10. There are approximately 75k training and 25k test images. We see in Table 1 a particularly pronounced difference in performance between BayesGAN and the alternatives. Data samples are shown in Figure 5.

## 4.5 CelebA

The large CelebA dataset contains 120k celebrity faces amongst a variety of backgrounds (100k training, 20k test images). To reduce background variations we used a standard face detector [12] to crop the faces into a standard $50 \times 50$ size. Figure 5 shows data samples from the trained Bayesian GAN. In order to assess performance for semi-supervised learning we created a 32-class classification task by predicting a 5-bit vector indicating whether or not the face: is blond, has glasses, is male, is pale and is young. Table 1 shows the same pattern of promising performance for CelebA.

# 5 Discussion

By exploring rich multimodal distributions over the weight parameters of the generator, the Bayesian GAN can capture a diverse set of complementary and interpretable representations of data. We have shown that such representations can enable state of the art performance on semi-supervised problems, using a simple inference procedure.

Effective semi-supervised learning of natural high dimensional data is crucial for reducing the dependency of deep learning on large labelled datasets. Often labeling data is not an option, or it comes at a high cost – be it through human labour or through expensive instrumentation (such as LIDAR for autonomous driving). Moreover, semi-supervised learning provides a practical and quantifiable mechanism to benchmark the many recent advances in unsupervised learning.

Although we use MCMC, in recent years variational approximations have been favoured for inference in Bayesian neural networks. However, the likelihood of a deep neural network can be broad with many shallow local optima. This is exactly the type of density which is amenable to a sampling based approach, which can explore a full posterior. Variational methods, by contrast, typically centre their approximation along a single mode and also provide an overly compact representation of that mode. Therefore in the future we may generally see advantages in following a sampling based approach in Bayesian deep learning. Aside from sampling, one could try to better accommodate the likelihood functions common to deep learning using more general divergence measures (for example based on the family of $\alpha$-divergences) instead of the KL divergence in variational methods, alongside more flexible proposal distributions.

In the future, one could also estimate the marginal likelihood of a probabilistic GAN, having integrated away distributions over the parameters. The marginal likelihood provides a natural utility function for automatically learning hyperparameters, and for performing principled quantifiable model comparison between different GAN architectures. It would also be interesting to consider the Bayesian GAN in conjunction with a non-parametric Bayesian deep learning framework, such as deep kernel learning [13, 14]. We hope that our work will help inspire continued exploration into Bayesian deep learning.

**Acknowledgements** We thank Pavel Izmailov and Ben Athiwaratkun for helping to create a tutorial for the codebase, helpful comments and validation. We also thank Soumith Chintala for helpful advice. We thank NSF IIS-1563887 for support.

## Footnotes

[1]For mini-batches, one must make sure the likelihood and prior are scaled appropriately. See Supplement.

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
