[Supplementary Material]

# Bayesian GAN

**Yunus Saatchi**
Uber AI Labs

**Andrew Gordon Wilson**
Cornell University

In this supplementary material, we provide (1) futher details of the MCMC updates, (2) illustrate a tutorial figure, (3) show data samples from the Bayesian GAN for SVHN, CIFAR-10, and CelebA, and (4) give performance results as a function of iteration.

## 1 Rescaling conditional posteriors to accommodate mini-batches

The key updates in Algorithm 1 involve iteratively computing $\log p(\theta_g|\mathbf{z}, \theta_d)$ and $\log p(\theta_d|\mathbf{z}, \mathbf{X}, \theta_g)$, or $\log p(\theta_d|\mathbf{z}, \mathbf{X}, \mathcal{D}_s, \theta_g)$ for the semi-supervised learning case (where we have defined the supervised dataset of size $N_s$ as $\mathcal{D}_s$). When Equations (1) and (2) are evaluated on a *minibatch* of data, it is necessary to scale the likelihood as follows:

$$\log p(\theta_g|\mathbf{z}, \theta_d) = \left( \sum_{i=1}^{n_g} \log D(G(\mathbf{z}^{(i)}; \theta_g); \theta_d) \right) \frac{N}{n_g} + \log p(\theta_g|\alpha_g) + \text{constant} \qquad (1)$$

For example, as the total number of training points $N$ increases, the likelihood should dominate the prior. The re-scaling of the conditional posterior over $\theta_d$, as well as the semi-supervised objectives, follow similarly.

## 2 Additional Results

Figure 3: We illustrate a multimodal posterior over the parameters of the generator. Each setting of these parameters corresponds to a different generative hypothesis for the data. We show here samples generated for two different settings of this weight vector, corresponding to different writing styles. The Bayesian GAN retains this whole distribution over parameters. By contrast, a standard GAN represents this whole distribution with a point estimate (analogous to a single maximum likelihood solution), missing potentially compelling explanations for the data.

Figure 4: Test accuracy as a function of iteration number. We can see that after about 10000 SG-HMC iterations, the sampler is mixing reasonably well. We also see that per iteration the Bayesian GAN with SG-HMC is learning the data distribution more efficiently than the alternatives.

## CIFAR10

## SVHN

## CelebA

Figure 5: Data samples for the CIFAR10, SVHN and CelebA datasets from four different generators created using four different samples from the posterior over $\theta_g$. Each panel corresponding to a different $\theta_g$ has different qualitative properties, showing the complementary nature of the different aspects of the distribution learned using a fully probabilistic approach.

Figure 6: A larger set of data samples for CIFAR10 from four different generators created using four different samples from the posterior over $\theta_g$. Each panel corresponding to a different $\theta_g$ has different qualitative properties, showing the complementary nature of the different aspects of the distribution learned using a fully probabilistic approach.

Figure 7: A larger set of data samples for SVHN from four different generators created using four different samples from the posterior over $\theta_g$. Each panel corresponding to a different $\theta_g$ has different qualitative properties, showing the complementary nature of the different aspects of the distribution learned using a fully probabilistic approach.

Figure 8: A larger set of data samples for CelebA from four different generators created using four different samples from the posterior over $\theta_g$. Each panel corresponding to a different $\theta_g$ has different qualitative properties, showing the complementary nature of the different aspects of the distribution learned using a fully probabilistic approach.