[Reviews · NeurIPS 2017]

Reviewer 1



Summary: The paper introduces a Bayesian type of GAN algorithms, where the generator G and discriminator D do not have any fixed initial set of weights that gets gradually optimised. Instead, the weights for G and for D get sampled from two distributions (one for each), and it is those distributions that get iteratively updated. Different weight realisations of G may thus generate images with different styles, corresponding to different modes in the dataset. This, and the regularisation effect of the priors on the weights, promotes diversity and alleviates the mode collapse issue. The many experiments conducted in the paper support these claims. Quality, Originality, Clarity: While most GAN papers today mainly focus on slight variations of the original net architecture or of the original GAN objective function, this paper really presents a new, original and very interesting algorithm that combines features of GANs with a strongly Bayesian viewpoint of Neural Nets (similar in spirit, for example, to the viewpoint conveyed by Radford Neal in his PhD thesis), where the weights are parameters that should be sampled or integrated out, like any other parameter in the Bayesian framework. This leads to a very elegant algorithm. The paper contains many good experiments that prove that the algorithm is highly effective in practice (see Fig 1 and Table 1), and the produced images (though not always state of the art it seems) indeed reflect a high diversity of the samples. Moreover, the paper is very clearly written, in very good english, with almost no typos, and the algorithm is well explained. Overall: a very good paper! Other remarks: - Am I right in understanding that at each \theta_g sampling step, you are actually not really sampling from p(\theta_g | \theta_d), but rather from something more like p(\theta_g | \theta_d) * p(theta_d | theta_g^old) (and similar for p(\theta_d | theta_g)) ? In that case, this should be stressed and explained more clearly. In particular, I got confused at first, because I did not understand how you could sample from p(\theta_g | \theta_d) if \theta_d was not fixed. - Algorithm 1: the inner loop over M does not stick out clearly. The 'append \theta_g^k to sample set' should be inside this loop, but seems outside. Please ensure that both inner- and outer loop are clearly visible from first eye sight. - l.189 & l.192: clash of notations between z (of dimension d, to generate the synthetic dataset) and z (of dimension 10, as random input to the generator G). Use different variable for the first z - Figure 1: What is p_{ML-GAN} ? I guess that the 'ML' stays for 'Maximum-Likelihood' (like the maximum-likelihood gan mentioned on l.267 without further introduction.) and that the MLGAN is just the regular GAN. In that case, please call it simply p_{GAN}, or clearly introduce the term Maximum-Likelihood GAN and its abbreviation MLGAN (with a short justification of why it is a maximum likelihood, which would actually be an interesting remark). - l.162-171: Describe the architectures more carefully. Anyone should be able to implement *exactly* your architecture to verify the experiments. So sentences like ('The networs we use are slightly different from [10] (e.g. we have 4 hidden layers and fewer filters per layer)', without saying exactly how many filters, is not ok. Also, the architecture for the synthetic dataset is described partly at l.162-163 and partly at l.191. Please group the description of the network in one place. The authors may consider describing roughly their architectures in the main part, and put a detailed description in the appendix. Response to rebuttal: We thank the authors for answering our questions. We are happy to hear that the code will be released (as there is usually no better way to actually reproduce the experiments with the same parameter settings). Please enclose the short explanations of the rebuttal on GANs being the Maximum Likelihood solution with improper priors into your final version.

Reviewer 2



This paper provides a Bayesian formulation for GAN. The BayesGAN marginalizes the posteriors over the weights of the generator and discriminator using SGHMC. The quantitive experiment result is state-of-art, although the sample quality is not impressive. The paper is well written and easy to follow. My concern is under relative low dimension output, the model seems is able to avoid model collapse. It is not clear for higher dimension output, whether Bayes GAN is still able to capture the distribution of data. Another issue is to my knowledge, SGHMC converges much slower than normal SGD with momentum. In senthetic dataset, BayesGAN converges faster, but it is not clear for complex input distribution, the BayesGAN is still keeping advantages in converge. I also suggest to change the legends of figures, because “ML GAN” is quite misleading.

Reviewer 3



Overview This work proposes a fully Bayesian approach to Generative Adversarial Nets (GANs) by defining priors over network weights (both the discriminator and the generator) and uses stochastic Gradient Hamiltonian Monte Carlo to sample from their posteriors. A fully Bayesian approach could allow for more extensive exploration of the state space and potentially avoid the mode collapse problem afflicting other GANs. Authors study this approach within the scope of semi-supervised learning and show that the proposed technique can achieve state of the art classification performance on several benchmark data sets but more importantly, can generate a more diversified artificial samples implying multimodal sampling behaviour for network weights. Quality The paper makes an important contribution toward closing the gap between purely data driven GANs susceptible to mode collapse problem and more interpretable Bayesian generative models that can produce diverse representations of data. Results of experiments with several benchmark and a synthetic data set are quite encouraging. Samples generated by the proposed model seem to be more diverse and expressive than those generated by competing GANs. Clarity It was a pleasure reading this paper. Very well written and organized. Originality Bayesian approach to GANs is taken by assigning Bayesian priors (albeit vague) over network weights and using SGHMC to sample from their posteriors. To the best of my knowledge this work is original. Significance The proposed work has the potential to open a new avenue for research in GANs and as such can be considered highly significant. Comments: - The posteriors for \theta_d and \theta_g are empirical approximations. Do we really know the sampler can still converge to the invariant distributions given that the conditional distributions are only crude approximations of the true distributions? Any analysis (theoretical or experimental) done? - Algorithm 1 runs for 5000 iterations and iterations are run sequential. What parts of this approach are parallelizable? How does the run time compares to standard GANs? - There has been some recent work in GANs that use finite Gaussian mixture models to generate samples in the code space (produced by a stacked denoising autoencoder). These approaches have similar motivation (to be able to generate multi mode and more representative data sets) as the current work but they approach the problem from a visualization perspective. It may be worthwhile to mention them. Typo: Line 188: a set generators should be a set of generators.